# Properties of Geopolymers Based on Metakaolin and Soda-Lime Waste Glass

**DOI:** 10.3390/ma16155392

**Published:** 2023-07-31

**Authors:** Galyna Kotsay, Paweł Grabowski

**Affiliations:** Faculty of Civil Engineering, Mechanics and Petrochemistry, Warsaw University of Technology, Lukasiewicza St. 17, 09-400 Plock, Poland; pawel.grabowski@pw.edu.pl

**Keywords:** soda-lime waste glass, geopolymer, paste, alkaline activity, dynamic viscosity, strength

## Abstract

The paper determines the properties of geopolymer pastes based on metakaolin and soda-lime waste glass. The density, alkaline activity, strength and microstructure of the reference geopolymer, as well as geopolymers with a 10%, 30% and 50% soda-lime waste glass content instead of metakaolin, were tested. The experimental results indicate that the properties of the geopolymers with waste glass largely depend on the ratio of the liquid to solid substance. Increasing the content of waste glass causes an increase in the fluidity of the geopolymer paste, which in turn allows the amount of water glass, i.e., an activator during the obtaining of geopolymers, to be reduced. On the basis of the conducted tests, it was found that the strength of geopolymers can be increased by adding up to 50% of soda-lime waste glass instead of metakaolin and by having a lower content of water glass.

## 1. Introduction

In the construction industry, innovative materials, in particular recycled materials, are becoming increasingly important. A very valuable material for recycling is glass waste. An important feature of glass is the possibility of it being repeatedly processed without its properties changing. Glass recycling saves natural resources, reduces carbon dioxide emissions into the atmosphere, and reduces the energy that is needed to produce new products [1,2]. According to the FEVE association (European Federation of Glass Packaging Manufacturers), in the European Union and Great Britain, 79% of glass was recycled in 2020 [3] and, by 2030, there is an aim to achieve a 90% recycling level of glass packaging. The glass industry produces more and more glass, with the amount of glass waste also increasing accordingly. Some of the waste is recovered and returned to the production process of glass products. However, the amount of glass that can be reused in production is limited due to the deterioration of the quality of the obtained products. Glass waste can also be used for the production of fiberglass, mineral wool, abrasives or road paints [4]. However, to date, glass waste accounts for 7 to 10% of all the waste that is deposited in landfills.

At present, more and more ways to reduce the amount of waste, as well as methods to reuse it in accordance with the principles of sustainable development, are being sought. There are many publications concerning the use of glass as an aggregate or non-clinker component for cement [5,6,7,8,9,10,11,12,13]. The use of glass as an aggregate for mortar or concrete reduces the strength of the obtained products as a result of the reduction in the adhesion between the slurry and the glass aggregate. Moreover, the use of glass as a non-clinker component of cement is also limited due to the high content of alkali in the waste glass, which in turn causes an alkali–silica reaction (ASR) [14,15,16,17,18,19,20]. The potential use of glass cullet as a non-clinker component of cement is not clear. Previous research [21,22,23,24,25] has proven that the use of ground glass instead of cement had no negative effects on the properties of cement after a long period of maturation. In turn, in [26,27,28,29,30], it was stated that the effect of glass on the expansion of the obtained products depends on the grain size of the ground glass. The use of glass with a grain size of less than 300 microns can reduce the impact of the expansion of ASR products. The authors of [31,32,33,34] proposed the use of waste glass as an alkaline activator in combination with other waste materials, e.g., blast furnace slag or fly ash.

When analysing the results of research concerning the possibility of using waste glass, the largest disadvantage of glass in building materials made from commonly used cement is the alkali content. This is why the use of glass cullet for geopolymers has been proposed in scientific research [35,36,37,38,39,40,41,42,43,44]. Geopolymers are considered to be an alternative material to cement and can be used for the production of prefabricated elements, structural elements, adhesive mortars or insulating materials [45,46,47]. It is derived from an active aluminosilicate material (called a precursor) and an activator. Metakaolin or industrial waste, such as fly ash, blast furnace slag or ceramic brick, are often used as precursors [48,49,50,51,52]. In turn, alkaline activators are used to dissolve precursors and to catalyse polycondensation reactions, as well as to stabilize aluminium cations in tetrahedral coordination in order to make the structure neutral. Water glass, NaOH or KOH bases, sodium or potassium sulphates, or carbonates are used as activators [46,53,54,55]. Water glass, which is used the most often, is considered to be the best activator because it introduces silicate ions into the solution. It is worth noting that water glass is an air binder, but the use of water glass as an activator, with a precursor, results in the obtaining of a geopolymer that is resistant to moisture after curing.

A properly designed and synthesized geopolymer has very good mechanical and physicochemical properties [56]. The binding and hardening mechanism of geopolymers significantly differs from that of ordinary cement. The cement setting process is based on the hydration reaction, whereas the geopolymer setting process is based on the polycondensation reaction [46]. As a result of polycondensation in the geopolymer, amorphous or semi-crystalline groups of silicon and aluminium oxide tetrahedra are formed, which are connected alternately by common oxygen atoms. The main difficulty in the synthesis of geopolymers is the variability of the chemical and mineralogical composition of the precursors. This results in a different activation process and a very diverse physicochemical structure of geopolymers. Therefore, there are no clear guidelines for the mechanism of geopolymerization.

Glass waste is a source of amorphous silica and alkalis, making them not only a precursor but also an activator in geopolymers. However, the potential use of glass waste as a substitute for conventional precursors is limited. In most studies, the optimal content of glass waste as a precursor is up to 15% [36,37,39,42]. Therefore, the aim of this research is to increase the content of soda-lime waste glass (SLWG) in metakaolin-based geopolymers.

## 2. Materials and Methods

Metakaolin produced by the firm Rominco Polska [57], soda-lime waste glass produced by REWA [58] and glass water produced by Rudniki S.A. [59] were used as the main object of the research. The chemical compositions and properties of the materials are shown in Table 1. The real density and the Blaine specific surface test of materials were performed according to the methods described in the standards [60,61]. Sodium hydroxide of 99% purity by [62] was used to reduce the modulus of the water glass to M = 1.5.

The rheological measurements (dynamic viscosity and fluidity) were conducted using the IKA ROTAVICS me-vi. The geopolymer mixture was prepared in a mixer. The mixing time was 3 min. The measurement of rheological parameters was conducted 15 min after the completion of mixing the components. The temperature during the test was maintained at a constant level of 20 °C. The reaction speed of the pastes was assessed on the basis of heat measurements. Calorimeter Calmetrix I-Cal 2000 HPC was applied to determine the effect of waste glass on the polycondensation reactions of metakaolin. The rate of heat evolution was recorded for the pastes with a liquid-to-binder ratio equal to 0.5. Solid mixtures were made from metakaolin and glass as partial metakaolin substitution. The amount of the heat released was monitored every 15 s for the 48 h. The rate of heat evolution was measured at the isothermal condition (20 °C).

The specimens of pastes were made in moulds sized 20 mm × 20 mm × 20 mm. All cuboids of pastes were taken out from the moulds after one day; then, part of the samples were cured at 20 °C and part of the samples were stored at the 40 °C and 80 °C for 24 h. The apparent density and the compressive strength tests were evaluated on a series of six specimens. Samples were examined using hydraulic press with the 500 N/s force increase rate. Apparent density tests were conducted according to [60]. Pieces of geopolymers after the compression test were used for microstructure observations by SEM using a ZEISS EVO 10 with an acceleration voltage of 10.0 kV.

To determine the alkaline activity of geopolymers, distilled water was used as an extractant with a ratio of the specimen surface to the extractant volume of 0.34 cm^−1^. The alkaline activity of the paste was determined on a series of six specimens. For the quantitative analysis of alkali content, a flame photometer FP902 (PG Instruments Limited, Alma Park, Wibtoft, Leicestershire, UK) with an accuracy of ±0.5% was used. The results of the alkaline activity of specimens are presented in units of ppm/m^2^ of paste.

## 3. Results and Discussion

The parameters of SiO_2_/Al_2_O_3_, Na_2_O/Al_2_O_3_ and Na_2_O/SiO_2_ are important when designing geopolymers. Geopolymers with a polysialate-syloxo structure are believed to have a higher strength and therefore, a molar ratio of SiO_2_/Al_2_O_3_ ranging from 3.3 to 4.5 is optimal [39,63,64]. In turn, the alkali content affects the solubility of aluminosilicates and the stabilization of aluminium cations in tetrahedral coordination. Thus, when obtaining geopolymers, Na_2_O/Al_2_O_3_ ranging from 0.8 to 1.6 and Na_2_O/SiO_2_ from 0.20 to 0.48 are optimal. Many publications describe the influence of some parameters on the properties of geopolymers [46,63,64], but to date, there is no clear answer regarding the relationship between a given module and these properties. Geopolymers are still seen as a modern material, despite the fact that they have been known for several decades.

In the research, two materials—metakaolin and glass waste—were used as the precursor, whereas water glass with a constant molar ratio of SiO_2_/Na_2_O = 1.5 was used as the activator. In the case of the reference geopolymer, 100% of metakaolinite was used as the precursor, and in the other samples, metakaolin was replaced with 10%, 30% and 50% of soda-lime waste glass. The liquid to solid ratio for the geopolymer reference, 10SLWG0.9, 30SLWG0.9 and 30SLWG0.9 samples, was 0.9. The composition of the samples is shown in Table 2.

In metakaolin, the molar ratio of SiO_2_/Al_2_O_3_ is 2.15, and the adding of an activator to metakaolin increases this parameter to 3.23. Replacing 10%, 30% and 50% metakaolin with SLWG that contains 70% SiO_2_ also increases the molar ratio of SiO_2_/Al_2_O_3_ from 3.53 to 7.15. Waste glass still contains 14% of alkali, and therefore the use of glass also acts as an activator and increases the Na_2_O/Al_2_O_3_ and Na_2_O/SiO_2_ parameters. The molar ratio of the designed geopolymers is shown in Table 2.

When comparing the molar ratio of the designed geopolymers to the optimal parameters (according to literature data [63,64]), the use of 10% of SLWG instead of metakaolin meets all the required parameters. In turn, the sample with 30% of SLWG only meets the parameters of Na_2_O/Al_2_O_3_ and Na_2_O/SiO_2_, whereas the sample with 50% of SLWG meets the parameters for only Na_2_O/SiO_2_.

It is known that the geopolymerization process depends on the type of precursor, the concentration of the activator, and the method of curing. In order to investigate how water glass affects the dissolution of the precursor, calorimetric tests were first performed for the “metakaolin and water glass” sample, and then for the “waste glass and water glass” sample. Glass with different alkalinity was used for the study.

When adding metakaolin to water glass, the first stage involves the adsorption of the water glass solution on the surface of the precursor, and then its dissolution, which is associated with the release of heat. Figure 1a shows that, the greater the alkalinity of the water glass (the molar ratio of Na_2_O/SiO_2_), the greater the solubility of the metakaolin. The released heat increases exponentially with an increasing alkalinity of the activator. In turn, the heat from the reaction of waste glass with water glass was ten times lower when compared to the metakaolin (Figure 1b). It was observed that, for a given sample, the molar ratio of the Na_2_O/SiO_2_ activator (within the range from 0.5 to 0.7) increased the release of heat, while in the case of higher alkalinity, no changes in heat were observed.

Although the use of waste glass instead of metakaolin in geopolymers increases the molar ratio of Na_2_O/Al_2_O_3_ and Na_2_O/SiO_2_, its activity, when compared to metakaolin, is ten times lower. Therefore, the introduction of waste glass instead of metakaolin into geopolymers causes a decrease in the reaction rate with an increase in the glass content (Figure 2). The introduction of soda-lime waste glass in the amount of 10% does not significantly change the heat of hydration, while the introduction of 30% and 50% instead of metakaolin reduces the total heat of hydration after 48 h to 24% and 38%, respectively.

The basic properties of geopolymers are their rheological properties. The use of a geopolymer with an appropriate dynamic viscosity allows the aggregate to be encapsulated, and a more homogeneous structure to be obtained. The viscosity results of the tested samples are presented in Table 3, in which they are compared to that of a cement paste with a standard consistency.

The research showed that the introduction of soda-lime waste glass instead of metakaolin significantly reduced the dynamic viscosity of the geopolymers when compared to the reference geopolymer. The addition of 10% of SLWG reduced the dynamic viscosity by five times, while the addition of more than 30% of soda-lime waste glass caused its viscosity to decrease significantly. The reduction in viscosity increased the fluidity of the geopolymer. The introduction of more than 30% of soda-lime waste glass caused fluidity to increase quickly, which in turn violated the continuity of the mixture’s structure and reduced the cohesive forces in the material. When compared to the cement paste of a standard consistency, the reference geopolymer had a much greater plastic viscosity. However, only when 10% to 30% of soda-lime waste glass was introduced, instead of metakaolin, was the fluidity comparable to that of the cement paste.

The effect of soda-lime waste glass and curing temperature on the bulk density is shown in Table 4. In all the samples, after 7 days of thermal curing, the density was reduced. This was associated with a decrease in the weight of the samples as a result of water evaporation during the polycondensation reaction. The reference geopolymer had the highest density due to the higher reactivity of metakaolin. Waste glass, when compared to metakaolin, has a lower specific surface and activity, and therefore the introduction of glass instead of metakaolin into geopolymers causes a decrease in their density.

Increasing the curing temperature increases the rate of the polycondensation of geopolymers, which in turn results in a decrease in their density. Thus, in the case of the reference sample, when curing at a temperature of 80 °C, there was a reduction in density of up to 8%. In turn, the thermal hardening of the samples with the addition of soda-lime waste glass was characterized by a greater reduction in density—even up to 15% for the samples with a content of 50% of waste glass. The decrease in density was associated with a greater water evaporation due to an increased porosity. Over time, after 28 days, when curing at 20 °C, a greater decrease in density was observed in the case of the reference sample and the sample with the addition of 10% of SLWG, whereas for the samples with a higher content of SLWG, the density was approximately the same as that obtained after 7 days. By increasing the curing temperature after 28 days, the reference sample (inversely) had a greater weight loss than the samples with the addition of SLWG. The greater weight loss after the hardening of the sample can cause cracking.

The influence of soda-lime waste glass and temperature curing on the compressive strength of geopolymers are presented in Figure 3. In the reference samples cured at 20 °C, the strength reached 32 MPa at 7 days and increased at 28 days to 51 MPa. The temperature increase in thermal curing accelerates the geopolymerization reaction; therefore, reference samples at 50 °C reached a higher strength at 7 days and 28 days (45 and 55 MPa, respectively). However, curing at 80 °C was observed to favour significant strength development in the early ages to 71 MPa but decreased mechanical strength to 63 MPa at 28 days; this reduction in strength was due to the formation of fissures caused by shrinkage.

Replacing metakaolin with 10% SLWG does not significantly change the strength of the samples stored at room temperature and thermal curing at 50 °C. A decrease in strength was observed for curing at 80 °C compared to the control samples. With 30% SLWG instead of metakaolin, the strength does not change only for samples stored at room temperature at 7 days. For the samples in question, no increase in strength was observed after 28 days. Increasing the content to 50% SLWG, significantly falls geopolymer strength after 7 days by 50% from the control sample values and strength falls after 28 days compared to strength result tests after 7 days.

The reduction in the strength of geopolymers over time is related to the alkalinity of the solution, which can cause the aluminosilicate gel to break down. It is known that an excess of alkali destroys Si–O–Si stronger bonds, forming Si–O–Na species. For the tested mixtures with 30% and 50% SLWG content, the compressive strength decreased due to the excess of Na^+^ ions in the mixtures. According to the results of the alkaline activity of the products, the amount of extracted sodium cations from a unit area increased 1.5 times for the system with 30% SLWG content and four times with 50% SLWG content (Table 5). However, the introduction of 10% SLWG instead of metakaolin slightly decreased the amount of extracted sodium and potassium ions.

Relevant studies were conducted on the cement paste to compare the amount of extracted alkaline ions; the results are presented in Table 5. The data show that the cement paste had a reverse extraction of alkali ions, more K^+^ extracted and the least amount of sodium ion compared to geopolymers. It should be noted that the reference geopolymers and 10 to 30% SLWG geopolymers have a lower total amount of extracted cations. Increasing the SLWG content in the geopolymer to 50% increases the overall alkalinity for geopolymer compositions.

The temperature increase in thermal curing accelerates the geopolymerization reaction, so the reference samples decreased total amount extracted. The total amount of extracted cations from the pastes’ surface vs. temperature geopolimiryzation are presented in Figure 4. The reduction in the alkali extraction was also noted for the geopolymer from 10% SLWG. An opposite situation is observed when increasing the waste glass content in geopolymer to 30%, as the temperature increase in thermal curing accelerates the increased total amount extracted, which proves that the alkali was not bound.

In order to reduce the amount of alkali in the samples, geopolymers with a lower content of water glass were formed. The composition of the samples and the molar ratio of the designed geopolymers are shown in Table 6.

The reduction in water glass in the samples with the SLWG reduced the molar ratios of SiO_2_/Al_2_O_3_, Na_2_O/Al_2_O_3_ and Na_2_O/SiO_2_ when compared to the samples with a constant ratio of liquid to substance (Table 6). When comparing the molar ratios of the geopolymers with a lower content of water glass to the optimal parameters (according to literature data), the use of 10% and 30% of SLWG instead of metakaolin meets all the required parameters, while the sample with 50% of SLWG meets only the Na_2_O/Al_2_O_3_ and Na_2_O/SiO_2_ parameters. For the tested systems, the density and amount of extracted alkalis, as well as the strength of the evaluated systems, were tested. The test results are presented in Table 7.

The reduction in water glass in the samples with the SLWG significantly changed the properties of the geopolymers. It is known that the polycondensation process of geopolymers is associated with the evaporation of water, and therefore the reduction in water glass in the samples with the SLWG caused an inverse increase in density. When the samples were cured at 80 °C, a slight decrease in their strength was observed for those with 10% of SLWG. However, for the samples with the SLWG content from 30 to 50%, and in the case of a lower liquid/solid ratio, an increase in strength was noted. The structure of geopolymers with a reduced content of water glass is more compacted, as can be seen from the results of testing the extracted sodium and potassium ions. Therefore, in the 30SLWG0.6 sample (Table 6), the amount of extracted alkalis halved when compared to the 30SLWG0.9 sample (Table 2). In turn, in the case of the 50SLWG0.5 sample, the reduction in extracted alkalis decreased by three times. Changes in the strength with regards to the content of water glass correlated with the microstructure of the samples of the obtained geopolymers. Figure 5 and Figure 6 show the microstructure of the geopolymers that were cured at room temperature.

The geopolymer polycondensation process is associated with water evaporation and can cause micro-cracks in products during curing. The SEM microscope tests confirm the influence of liquid/solid ratio on the microstructure of the tested pastes. In the geopolymer pastes with a higher liquid/solid ratio of 0.9 (samples: geopolymer reference, 10SLWG0.9; 30SLWG0.9 and 50SLWG0.9—Figure 5), microcracks, which were formed as a result of shrinkage of the products of the polycondensation reaction, were found. In addition, in the case of the 50SLWG0.9 sample with a soda-lime waste glass content of up to 50%, the structure was the least homogeneous and the least compact, which was also confirmed by the results of the tests on the viscosity and strength of the geopolymer paste. In turn, the pastes that contained a lower liquid/solid ratio of 0.8, 0.6 and 0.5 did not have microcracks and instead had a largely homogeneous gel structure (samples: 10SLWG0.8; 30SLWG0.6; 50SLWG0.5—Figure 6). However, in all samples with the soda-lime waste glass, particles were identified that did not undergo complete reaction and were surrounded by a geopolymeric matrix. This will be the subject of further research and publications.

## 4. Conclusions

This study determined the effect of soda-lime waste glass on the properties of geopolymer pastes based on metakaolinite. Based on the results of this study, the following conclusions may be drawn.

The viscosity tests revealed that the introduction of sodium–calcium waste glass instead of metakaolin significantly reduces the dynamic viscosity of geopolymers compared to the reference geopolymer. The decrease in viscosity enhances the fluidity of the geopolymer, thereby allowing for a reduction in the amount of water glass.

The reduction in water glass in the samples with soda-lime waste glass markedly altered the properties of the geopolymers. The structure of geopolymers with a diminished water glass content exhibited a higher compaction, as evidenced by the results of the extracted sodium and potassium ion testing.

Introducing 30% and 50% of soda-lime waste glass instead of metakaolin in samples with a liquid-to-solid ratio of 0.9 resulted in a decrease in the compressive strength of the geopolymers compared to the reference geopolymer. However, the geopolymer containing 30% and 50% soda-lime waste glass at a liquid-to-solid ratio of 0.6 and 0.5, respectively, exhibited the highest compressive strength compared to the reference geopolymer.

## Figures and Tables

**Figure 1 materials-16-05392-f001:**
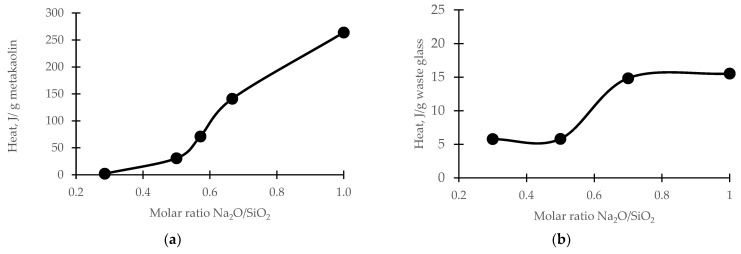
Calorimetric results of the reaction: (**a**) metakaolin with water glass; (**b**) waste glass with water glass.

**Figure 2 materials-16-05392-f002:**
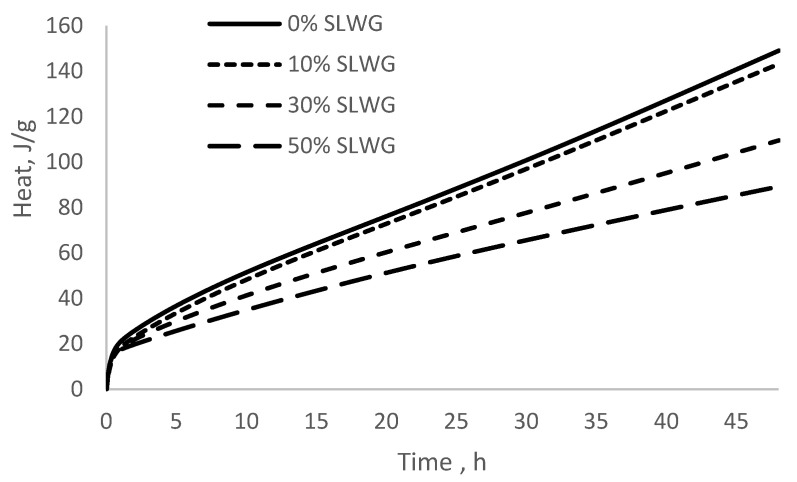
Heat of hydration for the samples with the addition of soda-lime waste glass (SLWG).

**Figure 3 materials-16-05392-f003:**
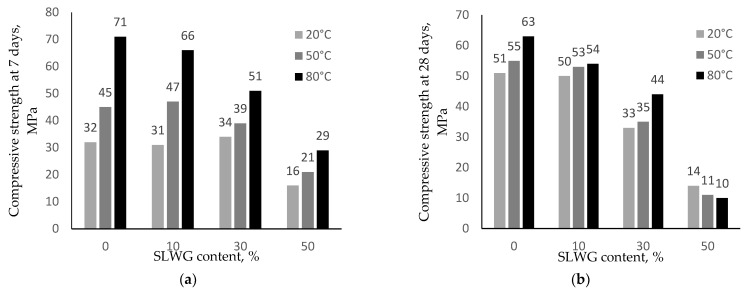
Compressive strength of the geopolymers at 7 days and 28 days.

**Figure 4 materials-16-05392-f004:**
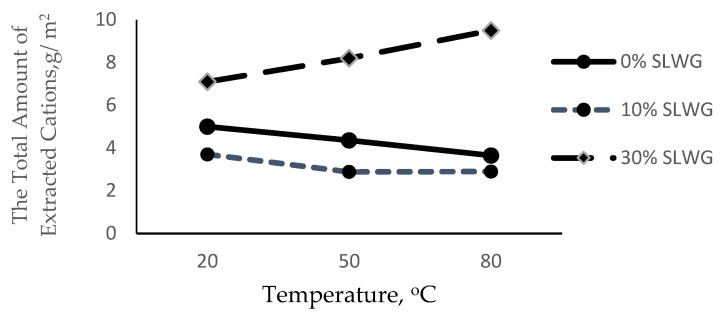
The total amount of extracted cations from the pastes’ surface vs. temperature geopolymerization.

**Figure 5 materials-16-05392-f005:**
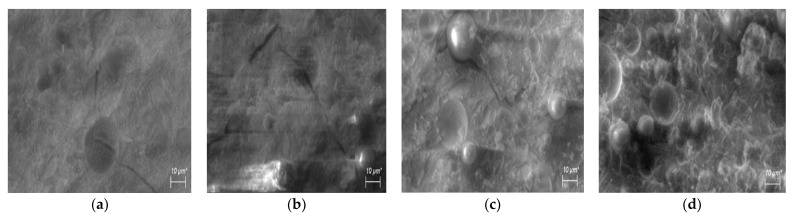
Microstructure for sample with a liquid/solid ratio of 0.9: (**a**) geopolymer reference; (**b**) 10SLWG0.9; (**c**) 30SLWG0.9; and (**d**) 50SLWG0.9.

**Figure 6 materials-16-05392-f006:**
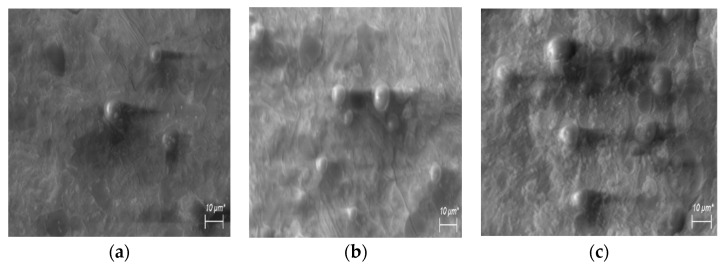
Microstructure for samples with a lower liquid/solid ratio: (**a**) 10SLWG0.8; (**b**) 30SLWG0.6; and (**c**) 50SLWG0.5.

**Table 1 materials-16-05392-t001:** The chemical composition and basic properties of metakaolin, soda-lime waste glass and water glass.

Materiałs	Oxides (wt %)								Density, g/cm^3^	Specific Surface, m^2^/g
	SiO_2_	Al_2_O_3_	CaO + MgO	Na_2_O_eq_	Fe_2_O_3_	TiO_2_	H_2_O	LOI		
Metakaolin	52	41	0.30	1.40	1.3	0.7	-	1.1	2.50	20,000
Soda-lime waste glass (SLWG)	72	1.0	12.0	14	-	-	-	-	2.43	3987
Sodium water glass (WG)	29	-	-	9	-	-	62	-	1.45	-

**Table 2 materials-16-05392-t002:** Mixture compositions and molar ratio of the prepared geopolymer pastes with a liquid/solid ratio of 0.9.

Sample Name	Mixture Composition (wt %)			Liquid/Solid	Molar Ratios		
	Metakaolin	SLWG	WG		SiO_2_/Al_2_O_3_	Na_2_O/Al_2_O_3_	Na_2_O/SiO_2_
Geopolymer reference	100	0	90	0.9	3.23	0.79	0.24
10SLWG0.9	90	10	90	0.9	3.53	0.93	0.26
30SLWG0.9	70	30	90	0.9	4.83	1.33	0.28
50SLWG0.9	50	50	90	0.9	7.15	2.03	0.28

**Table 3 materials-16-05392-t003:** Tests of viscosity in the case of the reference sample and the samples with 10%, 30% and 50% of SLWG.

Sample	Specifications		
	Density of Fresh Pastes, g/cm^3^	Dynamic Viscosity, Pa·s	Fluidity of Fresh Pastes, 1/Pa
Geopolymer reference	1.83	24.5	0.04
10SLWG0.9	1.84	5.57	0.18
30SLWG0.9	1.8	0.99	1.02
50SLWG09	1.72	0.14	6.94
Cement paste	2.06	2	0.5

**Table 4 materials-16-05392-t004:** Bulk density and the loss of density of the geopolymers at 7 days and 28 days.

Sample	Bulk Density after 7 Days, g/cm^3^		Loss of Density after 7 Days, %	Bulk Density after 28 Days, g/cm^3^		Loss of Density after 28 Days, %
	20 °C	80 °C		20 °C	80 °C	
Geopolymer reference	1.71	1.58	7.6	1.62	1.54	4.9
10SLWG0.9	1.71	1.48	13.5	1.62	1.51	4.4
30SLWG0.9	1.67	1.48	11.4	1.64	1.51	2.5
50SLWG0.9	1.6	1.36	15.0	1.62	1.62	0.6

**Table 5 materials-16-05392-t005:** The amount of extracted ions Na^+^ and K^+^ from the surface layers of the pastes.

Sample	Amount of Extracted, g/m^2^		The Total Amount of Extracted Cations, g/m^2^
	Na^+^	K^+^	
Geopolymer reference	4.7	0.32	5.0
10SLWG0.9	3.4	0.25	3.7
30SLWG0.9	6.79	0.35	7.1
50SLWG0.9	18.62	0.78	19.4
Cement paste	0.97	7.47	8.4

**Table 6 materials-16-05392-t006:** Mixture compositions and molar ratio of the prepared geopolymer pastes with a lower liquid/solid ratio.

Sample Name	Mixture Composition (wt %)			Liquid/Solid	Molar Ratios		
	Metakaolin	SLWG	WG		SiO_2_/Al_2_O_3_	Na_2_O/Al_2_O_3_	Na_2_O/SiO_2_
Geopolymer reference	100	0.0	90	0.9	3.23	0.79	0.24
10SLWG0.8	90	10	80	0.8	3.53	0.84	0.24
30SLWG0.6	70	30	60	0.6	4.37	0.99	0.23
50SLWG0.5	50	50	50	0.5	6.04	1.40	0.23

**Table 7 materials-16-05392-t007:** Tests results in the case of the reference sample and the samples with 10%, 30% and 50% of SLWG.

Sample	Bulk Density after 7 Days, g/cm^3^		Compressive Strength, MPa		Amount of Extracted, g/m^2^	
	20 °C	80 °C	20 °C	80 °C	Na^+^	K^+^
Geopolymer reference	1.71	1.54	32.23	71	4.7	0.32
10SLWG0.8	1.76	1.62	36.95	64.66	3.24	0.2
30SLWG0.6	1.77	1.58	45.00	73.82	2.9	0.22
50SLWG0.5	1.78	1.68	48.35	76.9	6.16	0.24

## Data Availability

The data presented in this study are available upon request from the corresponding author.

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
