# Peer review of "Properties of Geopolymers Based on Metakaolin and Soda-Lime Waste Glass"

_materials, 2023, doi:10.3390/ma16155392_

Round 1

Reviewer 1 Report

Based on the overall quality of this study, a major revision is needed.

Comments:

(1) What is the research gap of your study? Please clarify it in the introduction.

(2) The objective of this study is unclear. Please clarify it after the research gap in the introduction.

(3) The introduction is not comprehensive. More recent studies about the recycling of waste glass should be added to the introduction and discussed. For example: 1. Influence of size effect on the properties of slag and waste glass-based geopolymer paste; 2. A state-of-the-art review of crushed urban waste glass used in OPC and AAMs (geopolymer): Progress and challenges; 3. Influence of waste glass powder as a supplementary cementitious material (SCM) on physical and mechanical properties of cement paste under high temperatures. These studies can provide a more comprehensive literature review. And the background of your study can be strengthened after discussing these studies.

(4) For the geopolymer, the authors have to tell the difference between the geopolymer and alkali-activated materials (AAMs) in the introduction. For more information, the authors can learn from studies 1. 

Durability and microstructural characteristics of alkali activated materials made with waste glass as precursor: A review; 2. Fresh and mechanical properties overview of alkali-activated materials made with glass powder as precursor; 3. Geopolymer concrete as a cleaner construction material: An overview on materials and structural performances.

(5) For the figure 6, more in-depth discussions should be added to study the micro structures of the samples and clarify the inner mechanism.

(6) The conclusions are too simple. Please summarize your findings and results one by one.

The language should be polished.

Author Response

Dear Review,

Thank you very much for reviewing our manuscript.

Reviewer 2 Report

The article entitled “The Effect Of Waste Glass On The Properties Of Geopolymer” has been evaluated.  Authors have investigated the possibility of using as soda-lime waste glass as partial replacement of metakaolin during the synthesis of geopolymer materials. The authors investigated, among other things, the effect of curing temperature and water glass content on the resulting properties of geopolymers.

The article is very interesting and readable, but there are issues that need to be corrected (described in more detail below). However, I see potential in the article, and recommend a Major Revision before accepting.

Detailed comments:

Generally: the style of the tables and designation of materials and geopolymer product are inconsistent, units are given differently e.g. wt%/ wt % - please standardize.

Title: 

The title is not entirely accurate, because in your work you studied not only the effect of the waste glass content, but also the curing conditions and the water glass content of the mixture. I recommend changing/refining it but it is not a condition you have to fulfill. In the future, it may affect the citation of your paper because the title contains only one parameter studied. (Possible titles in my view: Geopolymers based on Waste Glass: Effect of mix design and curing temperature/ Properties of Geopolymers based on Waste Glass: Effect of mix design and curing temperature/ Geopolymers based on Waste Glass: Effect of mix design and curing temperature on properties/ etc.)

Keywords:

The last one (strength) has a bigger font.

Introduction:

“More and more” used in first and second paragraph is not very scientific. Try to find a suitable synonym for at least one of them.

You mention what percentage of the glass is recycled, but you don't say what quantity. Please add it.

Page 2, line 60-61. This sentence about possible activators should be supported by references.

Page 3 - it's a blank page, I don't know if it's an error in the manuscript or in the submission system.

Materials and Method:

Table 1: In the caption, there is a different designation of materials described in comparison with table and the rest of the text – container glasses/ soda-lime waste glass/waste glass; glass water/water glass. I think it would be helpful to define the acronym SLWG (soda-lime waste glass) and then use it in text, tables and figures.

In the table, titles “specific surface” and “density” are unreadable; there is mistake in “oxides” (oxydes); for the Al2O3 content of soda-lime waste glass, a decimal point is used instead of a dot.

Line 94: “…in a mixer. the mixing time…” change “t” to “T”.

Line 116: instead of +/- 0.5 % use a symbol ±.

Result and discussion:

 Table 2 and Table 6:  Instead of the first column N, I recommend to use the designation/sample designation/labels and use the designation you use in the text and tables below (e.g. 10%waste glass+90% WG). These designations/compositions (according to the tables) are not defined anywhere and the readers will have a problem with how you came to this.

In Table 2 and Table 6, mixture composition is in “g” and “mas%, respectively. Why? Please unify.

Line 144: “according to literature data” – which? Please add references.

Figure 1 and corresponding text: It's not entirely clear from my point of view. From the text I understand that Figure 1a shows the calometric results of the reaction of metakaolin and water glass and Figure 1b shows the reaction of waste glass and water glass. However, from the caption of the figure, it appears that 1a shows the results of the reaction of metakaolin and waste glass and 1b of metakaolin and water glass. In Figure 1a, the y-axis shows metakaolinit - change this to metakaolin to be consistent with the text. Please revise the text, graphs and captions.

Figure 2: The label refers to the results of the samples with the addition of waste glass, but the pure geopolymer (hereafter referred to as the geopolymer reference/reference sample in the tables) is also shown. Specify the figure caption.

Table 3, 4, 5, 7 – please correct the spelling error in the word geopolymer (geopolimer)

Line 205: “… when curing at 20°C.” In Section 2, Materials and Methods, you state that the samples were left at normal temperature. Is this temperature 20°C, if so, add it to section 2.

Table 4, 7:  Why are the data for curing at 40°C not mentioned?

Figure 4: Please change “oC” to “°C” in both graphs.

Line 243: K+ should have + as superscript.

Line 251: After “extracted” there is an extra space before the dot.

Figure 5: Please correct “geopolimiryzation” to “geopolymerization” in figure caption.

Figure 6: Images have not the same dimension, the scale is hard to see, for N1, there is the spelling error in the word geopolymer (geopolimer), for N2-N7 different designation is used in comparison with text - 10% glass+90% WG. I propose to split the images shown in Figure 6 into two - Figure 6 (formerly Figure 6a) and Figure 7(formerly Figure 6b). I would provide reference material for both images, but the editor may have a different opinion.

Conclusion: The conclusion must be completely rewritten. It does not summaries the results found. More information on the results found is in the abstract.

Author Response

(The authors gave the same response as above.)

Round 2

Reviewer 1 Report

The authors did not address my questions.

Author Response

Dear Review,

Thank you very much for the additional comments and suggestions.

We have modified the manuscript according to the comments.

Reviewer 2 Report

The review of article “The Effect Of Waste Glass On The Properties Of Geopolymer” (materials-2481054):

The article has been modified according to the comments or the authors have sufficiently defended their point of view. The changes really improved the article. I believe that after the adjustments have been made, the article is much more readable and has a much higher level.

However, the following still needs to be changed:

1)     Keywords: there's an extra space after soda-lime waste glass.

2)     I would define the soda-lime waste glass abbreviation on line 84, but this change is not necessary.

3)     Line 91, Table caption – please change “glass water” to “water glass” and “container glasses“ to “soda-lime waste glass“

4)     Line 130 – please change “SiO2/Na2O=1,5“to „SiO2/Na2O=1.5“

5)     In Figure 1a, the y-axis shows metakaolinit - change this to metakaolin to be consistent with the text. (mentioned in previous revision)

6)     Line 303 – “m” is missing in “icrocracks“

7)     Line 144/145: “according to literature data” – it is good that the references are mentioned in the introduction, but it would be better if the corresponding references were also mentioned here.

My conclusion is that the paper could be accepted after minor revision before publishing. My further revision after modifications is not necessary.

Author Response

Dear Review,

Thank you very much for the additional comments.
